# Current Situation and Utilization of Velvet Deer Germplasm Resources in China

**DOI:** 10.3390/ani12243529

**Published:** 2022-12-14

**Authors:** Lixin Tang, Xiaobin Wen, Ranran Zhang, Xiumei Xing

**Affiliations:** 1State Key Laboratory for Molecular Biology of Special Economic Animals, Institute of Special Animal and Plant Sciences of Chinese Academy of Agricultural Sciences, Changchun 130112, China; 2State Key Laboratory of Animal Nutrition, Institute of Animal Science, Chinese Academy of Agricultural Sciences, Beijing 100193, China

**Keywords:** velvet deer, germplasm resources, utilization

## Abstract

**Simple Summary:**

Livestock and poultry germplasm resources are national strategic resources related to the global revitalization of the seed industry. As an important part of livestock, velvet deer germplasm resources must be taken seriously. Our review focuses on the situation and utilization of velvet deer germplasm resources in China. We mainly introduce the situation of velvet deer germplasm resources in China, such as ecological distribution and the number of subspecies. We further elaborate on the domestication and breeding development of velvet deer in China and put forward suggestions on the scientific utilization of velvet deer germplasm resources.

**Abstract:**

Velvet deer are not only a representative special economic animal but also an important part of livestock. With the increasing awareness of international competition for germplasm resources in China, more and more attention has been paid to the protection and utilization of germplasm resources. However, there is poor understanding about velvet deer resources. Therefore, we are providing a comprehensive introduction of Chinese velvet deer germplasm resources from the aspects of ecological distribution, domestication and breeding.

## 1. Introduction

There are abundant deer resources in China, among which sika deer and red deer, the main velvet deer, have important economic value. Nowadays, due to the imbalance between supply and demand, international competition and other aspects, the velvet deer industry is developing slowly and the output value decreases yearly. Most importantly, weak awareness of velvet deer germplasm resource protection is also a direct cause of the slow development of the velvet deer industry that is resulting in its gradual loss of competitiveness. This is a reminder that scientific protection and rational utilization of velvet deer germplasm resources are the key to promoting the stable development of the velvet deer industry. In this paper, the present situation of velvet deer germplasm resources in China is introduced, the development of velvet deer germplasm resource utilization is expounded, and the scientific utilization of velvet deer germplasm resources is put forward, with the aim of providing the theoretical basis for velvet deer and other livestock germplasm resources protection and breeding innovation.

## 2. Present Situation of Velvet Deer Germplasm Resources

China is rich in deer resources, with 21 species, which accounts for 41.7% of deer species in the world [1]. Sika deer and red deer are two species belonging to the same cervus. They are the main velvet deer species in China because of their superior pilose antler production performance [2]. They are widely distributed in China. Sika deer are distributed in the northeast and south, while red deer are mainly distributed in the northwest (Figure 1).

### 2.1. Present Situation of Wild Sika Deer Germplasm Resources

There are six subspecies of sika deer in China, namely *Cervus nippon taiouanus, Cervus nippon hortulorum, Cervus nippon grassianus, Cervus nippon kopschi, Cervus nippon mandrinus* and *Cervus nippon sinchuanicus* [3]. Due to the change in environmental climate and the interference of humans, the number of wild sika deer has dramatically decreased. The existing subspecies of sika deer are *Cervus nippon hortulorum, Cervus nippon kopshi, Cervus nippon taiouanus* and *Cervus nippon sinchuanicus* [1]. *Cervus nippon hortulorum* are distributed in Heilongjiang and Jilin provinces. Because of excessive domestication, wild individuals were once considered extinct. Later, a small number were found in the border areas of southeast Heilongjiang province and eastern Jilin province [4]. *Cervus nippon kopshi* are mainly distributed in Anhui, Zhejiang and Jiangxi provinces [5]. The wild population of *Cervus nippon taiouanus* was recovered from extinction through artificial rearing. The number of wild individuals was about 1000 by 2013, and they mainly lived in Taiwan [6]. *Cervus nippon sinchuanicus* are mainly distributed in Sichuan Tiebu Sike Deer Nature Reserve, and the number reached about 800 in 1999 (Table 1) [7]. Wild populations of sika deer were listed as a first-class national protected animal in the latest released version of “The Lists of Wild Animals under Special State Protection in China (WASPC)” in 2021 [8]. In order to protect the germplasm resources and prevent the wild sika deer population from extinction, the habitats of the wild sika deer are mainly distributed in Chinese national nature reserves. This method has achieved remarkable results in the recovery of the wild sika deer. Taking Taohongling National Nature Reserve in Jiangxi Province as an example, the number of South China sika deer increased from 60 to 624 in 2021 [9].

### 2.2. Present Situation of Wild Red Deer Germplasm Resources

China has the most subspecies of red deer in the world. There are eight subspecies: *Cervus elaphus xanthopygus, Cervus elaphus wallichi, Cervus elaphus yarkandensis, Cervus elaphus songaricus, Cervus elaphus sibiricus, Cervus elaphus kansuensis, Cervus elaphus macneilli* and *Cervus elaphus alashanicus* [10]. Red deer are distributed in Xinjiang, Sichuan, Gansu, Ningxia, Tibet, Inner Mongolia, Heilongjiang and other places in China (Table 2). The habitat diversity of red deer is due to their strong adaptability to the environment. *Cervus elaphus yarkandensis* live in a droughty and hot environment of 40 °C. *Cervus elaphus wallichi* and *Cervus elaphus macnerlli* are distributed at high altitudes above 3800 m. *Cervus elaphus xanthopygus* can live in coniferous broad-leaved forests at −30−40 °C.

### 2.3. Current Situation of Domestic Sika Deer Germplasm Resources

Velvet deer have been domesticated in China for hundreds of years. At present, domestic sika deer in China are mainly domesticated from *C. n. hortulorum.* In addition, one local variety (Jilin sika deer), seven bred varieties (Shuangyang sika deer, Xifeng sika deer, Aodong sika deer, Siping sika deer, Xingkai lake sika deer, Dongda sika deer and Dongfeng sika deer) and a bred strain (Changbai mountain sika deer) have been bred.

Concerning the number of domestic sika deer, the available national census data show that by the end of 2004, there were 9465 farms in China and the total stock of sika deer was 452,355, of which, Jilin Province accounted for more than half of the national total stock [19]. In recent years, the number of domestic sika deer in China has not been surveyed nationwide. However, according to incomplete statistics, the total stock of domestic sika deer in China was about 500,000 by the end of 2018, with rearing in Shaanxi [20], Jilin [21], Hubei [22], Gansu [23], Jiangxi [24] and Yunnan [25], and the scale of rearing is constantly expanding.

### 2.4. Current Situation of Domestic Red Deer Germplasm Resources

Red deer are a kind of large-scale deer and have high production performance. The mainly domestic red deer are *Cervus elaphus songaricus*, *Cervus elaphus yarkandensis* and *Cervus elaphus xanthopygus*. *Cervus elaphus yarkandensis* have been cultivated on a large scale in Xinjiang, and the bred variety is called Tahe red deer. By April 2006, there were about 70,000 domestic red deer in Xinjiang [26]. Among them, there were 15,000 Tianshan red deer, 500–700 Altai red deer [26] and 54,000 Tahe red deer [27]. In addition, there is an improved variety named Qingyuan red deer in Liaoning Province [28], and the stock in 2003 reached about 15,000 [29].

### 2.5. Current Situation of Other Kinds of Velvet Deer Germplasm Resources

Germplasm resources, also known as genetic resources, generally refer to various biological types for specific germplasm or genes that can be used for breeding and other research. In addition to the wild and domestic living germplasm resources, the institute of Special Animal and Plant Science of the Chinese Academy of Agricultural Sciences has also established a comprehensive velvet deer germplasm resource pool, which contains a total of 4500 sika deer living resources, 7331 frozen semen samples, 150 tissue samples, 180 somatic cell samples and nearly 39,000 blood and DNA samples. In addition, more than 10,000 gene sequences (genome, transcriptome and proteome) have been preserved, and the chromosome-level genome of the sika deer has been assembled. Further, about 18,000 red deer blood samples and about 30,000 frozen semen samples have been collected. The velvet deer germplasm resource bank will provide sufficient material for the innovation of velvet deer germplasm resources.

## 3. Utilization of Velvet Deer Germplasm Resources

Germplasm resources are the material basis of modern breeding. Different types of germplasm resources have different genetic characteristics. An in-depth study of velvet deer germplasm resources will not only help to clarify the origin, evolution, classification and other issues, but it will also lay a great foundation for breeding innovation.

### 3.1. Application of Velvet Deer Germplasm Resources in Traditional Breeding

#### 3.1.1. Pure Breeding

As a basic breeding method, pure breeding plays a role in maintaining and improving the good quality of varieties. It has been widely used in the breeding of pigs [30], sheep [31], cattle [32] and other domestic animals. Shuangyang sika deer are the first velvet-used variety in China. They have been selected and bred through large-scale pure breeding for 23 years and have been introduced around China [33]. In the 1960s, the average velvet antler yield of Shuangyang sika deer exceeded 1 kg, ranking first in antler production in China [34].

#### 3.1.2. Cross-Breeding

Nowadays, antler velvet is a major source of income for the Chinese velvet deer industry. Therefore, velvet antler production performance has become the main goal of velvet deer breeding. As an important breeding method, hybridization is widely used in the modern livestock breeding industry and aims to breed new varieties with high production performance and improve the production efficiency of livestock, thereby achieving improved industrial gains. In order to improve the velvet antler-producing performance of velvet deer, China has carried out more than 70 years of cross-breeding with nearly 30 kinds of cross combinations, including hybridizations between red deer subspecies, sika deer varieties and interspecies hybridizations of sika deer and red deer [35].

Because of reproductive isolation, the offspring of most interspecific hybrids are not as adaptable as their parents. They are sterile or even lethal [36]. However, there is barely any hybridization incompatibility among velvet deer interspecies, which provides a feasible basis for variety improvement and breeding innovation of velvet deer. The characteristics of sika deer are relatively stable genetic performance, and the quality of their velvet antlers is better than that of red deer [37]. Sika deer are small in size and low in pilose antler yield, while red deer are large in size and high in pilose antler yield. Therefore, hybridization between sika deer and red deer has always been the main breeding method to improve production performance and breed new varieties. The physical appearance of F1s (Figure 2C,D) lies between the sika deer (Figure 2B) and the red deer (Figure 2A). F1s have normal reproductive capacity, the fertility and the performance of antler-producing reflect the great economic heterosis [38]. In addition, the growth performance of F1s, such as dressing percentage, carcass weight and net meat weight [39], show extremely significant heterosis [40]. The cytogenetic basis for the fertility of interspecific hybridization in velvet deer is the formation of a cis-trivalent structure of the telocentric/centromere chromosome during the meiosis of F1, which may be related to the balanced distribution of gametes [41]. The testis sections of F1s also show normal morphology and viable sperm; the testes are fully developed [42]. The fertility of the interspecific hybridization greatly improves the breeding innovation efficiency of the velvet deer.

### 3.2. Application of Velvet Deer Germplasm Resources in Molecular Breeding

#### 3.2.1. Molecular Assisted Breeding

With the development of biotechnology, the breeding of velvet deer has gradually developed to the molecular level. More and more molecular methods have been used to develop the utilization of velvet deer germplasm resources. The premise of the breeding innovation of velvet deer is to ensure the purity of the germplasm resources. The 1K chip, developed based on resequencing technology, can accurately distinguish sika deer, red deer and hybrids and judge the degree of hybridization [43]. Similarly, expression sequence labeling–simple sequence repeat (EST-SSR) is also used for identifying hybrids [44]. Based on the chip and EST-SSR, the parent individuals for breeding innovations can be scientifically selected. In addition, the unclear genetic relationships of velvet deer were also obstacles in the process of breeding innovation. Microsatellite markers are considered to be the first choice for characterizing the genetic diversity of populations [45], and they have been widely used for identifying the genetic relationship of species [46]. Ten STRs with high polymorphism have been developed to identify the genetic relationship among sika deer [47].

Molecular breeding of velvet deer is mainly based on molecular markers. Previous studies have found that ANXA2 [48], TRPC6 [49], HMG20A [49], and MTNR1A [50] are associated with velvet antler production performance. They also may become the candidate genes for screening paternal individuals during breeding. Transcriptome-based EST microsatellite markers have also been developed as important DNA markers for the selection of velvet antler producing performance [49,51].

#### 3.2.2. Genomic Selection

Genomic selection (GS) is a molecular breeding technology that uses the whole genome’s genetic markers of reference groups with genotypes and phenotypes to build models and predict the phenotypic value of genotype-only individuals [52]. It has become an important method for improving the economic traits of livestock and poultry due to its high accuracy, good character-selection effect and reductions in the economic and time costs of breeding. The application of genomic selection cannot be separated from the popularization of whole-genome-sequencing technology. Obtaining a high-quality whole genome of a species is the prerequisite for genomic selection. The high-quality chromosome-level genome of sika deer has been published [53] and used as a reference genome for a genome-wide association study(GWAS). A total of 94 SNPs significantly related to velvet antler production performance have been obtained through genomic selection; they are located in the exon regions of *OAS2* and *ALYREF/THOC4* [54], and it is speculated that *OAS2* and *ALYREF/THOC4* may be related to the weight of velvet antlers [55]. As a member of Cervidae with special evolutionary status, the chromosome-level genome of Tarim red deer has also been published [56]. However, it was mainly used for study on adaptive evolution. There are no reports on genomic selection and molecular breeding of red deer.

## 4. Suggestions for Conservation and Utilization of Velvet Deer Germplasm Resources in China

### 4.1. Completing the Identification and Evaluation System of Velvet Deer Germplasm Resources

The germplasm resources of velvet deer have rich diversity in phenotypes and genetic background. Therefore, multi-dimensional identification and evaluation should be conducted when identifying the germplasm resources of velvet deer. The appearance and the production performance of velvet deer should be evaluated according to the *National Standards for Chinese Sika Deer* and the *National Standards for Northeast Red Deer*. The purity of velvet deer should be detected by genome-sequencing technology. A genetic resource databank of velvet deer should be established. Velvet deer which have been identified for breeding should be registered, which would provide the basis for the follow-up work on germplasm resource protection and breeding innovation.

### 4.2. Establishing the Protection System of Velvet Deer Germplasm Resources

Breeding innovation with a stable genetic basis must be based on pure-breds. Cross breeding is an effective way of breeding innovation. Therefore, it is important to keep the balance between pure-bred resource protection and the development of breeding innovation. It is necessary to establish a sound system to protect the velvet deer germplasm resources as well as to make innovation in cross breeding to the greatest extent. The national sika deer conservation farm has been established to protect the local variety of sika deer in Jilin Province. The cooperation between research institutions and enterprises that rear sika deer has been established to lay a foundation for genetic improvement and breeding innovation of sika deer. The gene pool of velvet deer has also been created, which contains the genome, transcriptome and proteome data of velvet deer. It is a huge investment to construct national conservation farms. Therefore, national conservation farms can be built by relying on the large and medium-sized rearing enterprises of velvet deer, which cannot only reduce the cost of conservation farm construction but also maximize the use of velvet deer germplasm resources and improve the awareness of enterprises on the protection of velvet deer germplasm resources. In addition, the genetic resources of velvet deer should also be paid attention to. With the help of biotechnology, the characteristics of velvet deer genetic resources should be evaluated and the genes related to production performance can be identified to promote the development and utilization of velvet deer germplasm resources.

### 4.3. Accelerating the Innovation Process of Velvet Deer Germplasm

There have been nearly 75 years of cross-breeding of velvet deer in China. However, because of the lack of systematic and scientific selection, a new variety with excellent production performance and stable inheritance has not been obtained. It is urgent to speed up the process of velvet deer breeding innovation. The innovation of velvet deer germplasm resources requires more breeding of new varieties in addition to genetic improvement of existing velvet deer varieties. With the rapid development of biotechnology, molecular methods for breeding have been used on more and more livestock. Breeding based on species genome will be an inevitable choice for future breeding innovation. Therefore, a genomic selection system and high-density SNP breeding chips for different breeding needs should be developed to complete an efficient and scientific genome-based breeding system of velvet deer.

## 5. Conclusions

As a national strategic resource, livestock and poultry germplasm resources in China are facing fierce international competition. Velvet deer, as a special part of livestock, also face the same problem. In order to effectively solve the problem, an identification and evaluation system of velvet deer germplasm resources must be completed, a protection system of velvet deer germplasm resources must be established, and an innovation process of velvet deer germplasm resources must be accelerated so that the level of protection and utilization of velvet deer germplasm resources can be comprehensively improved in China.

## Figures and Tables

**Figure 1 animals-12-03529-f001:**
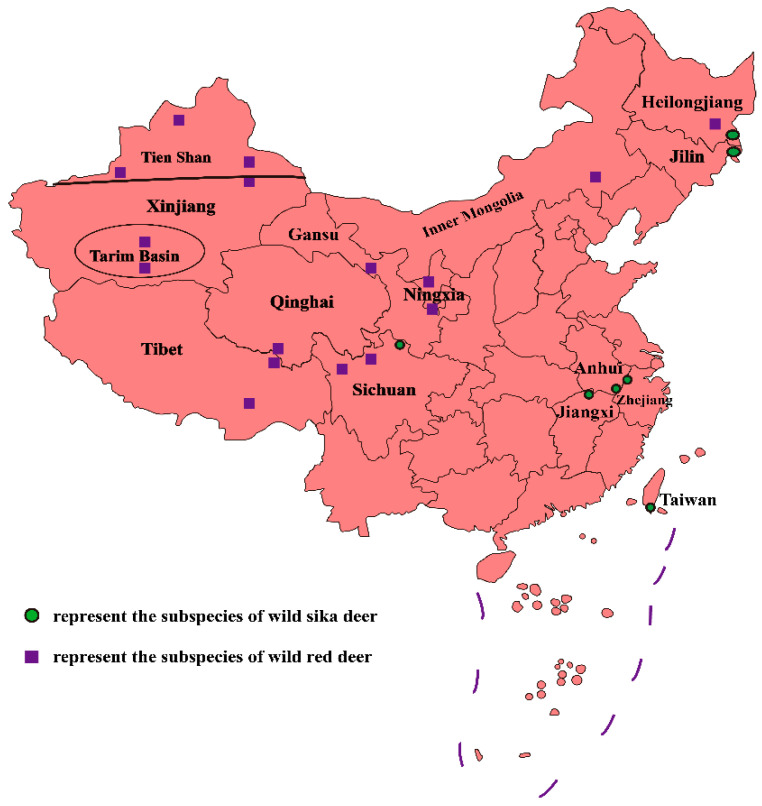
Distribution landscape of wild velvet deer in China. Green circles represent the distribution of subspecies of wild sika deer and purple squares represent the distribution of subspecies of wild red deer.

**Figure 2 animals-12-03529-f002:**
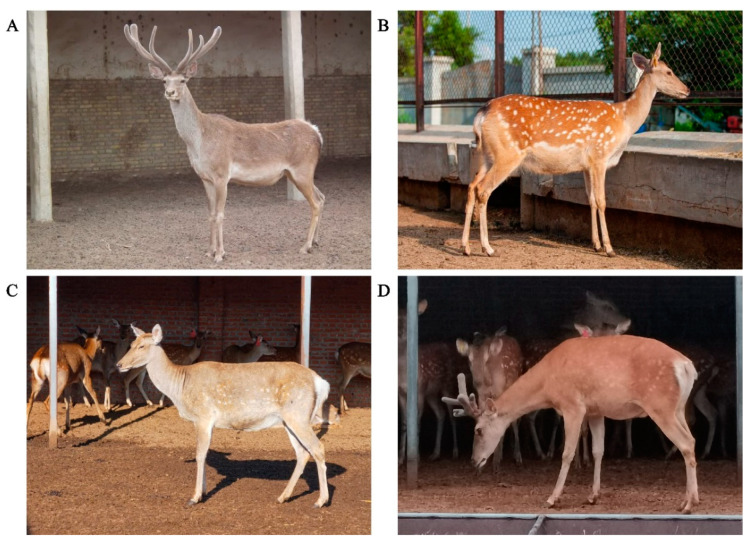
Appearance of F1s and their parents: (**A**) male red deer and (**B**) female sika deer; (**C**,**D**) F1 offspring.

**Table 1 animals-12-03529-t001:** Distribution and number of subspecies of wild sika deer.

Subspecies of Sika Deer	Distribution	Number
*Cervus nippon hortulorum*	Heilongjiang, Jilin province	---
*Cervus nippon kopshi*	Qingliangfeng National Nature Reserve, Taohongling National Nature Reserve and south of Anhui	More than 874
*Cervus nippon sinchuanicus*	Sichuan Tiebu Sike Deer Nature Reserve	About 800
*Cervus nippon taiouanus*	Kenting National Park, Taiwan	About 1000

**Table 2 animals-12-03529-t002:** Distribution and number of subspecies of wild red deer.

Subspecies of Red Deer	Distribution	Number
*Cervus elaphus wallichi* [11]	Sangri County, Tibet	About 300
*Cervus elaphus macneilli* [1]	Aba and Ganzi, Sichuan Province	---
*Cervus elaphus kansuensis* [12]	Qilian Mountain, Gansu, Qinhai, Ningxia, Southwest Sichuan and East Tibet	---
*Cervus elaphus yarkandensis* [13]	Northern Tarim Basin and desert of the southern plain, Xinjiang	About 450
*Cervus elaphus songaricus* [14]	Urumqi Nanshan, Hami Mountain and northern Tianshan Mountains, Xinjiang	Fewer than 10,000
*Cervus elaphus alashanicus* [15]	Helan Mountain, Ningxia	---
*Cervus elaphus sibiricus* [16]	Western and northern Xinjiang, Tianshan Mountain and forest and grassland in Altay	About 30,000
*Cervus elaphus xanthopygus* [17,18]	Saihanwula Nature Reserve, Inner Mongolia and Muling forest area, Heilongjiang	---

## Data Availability

Not applicable.

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
