# Peer review of "Current Situation and Utilization of Velvet Deer Germplasm Resources in China"

_animals, 2022, doi:10.3390/ani12243529_

Round 1

Reviewer 1 Report

This scientific manuscript adds a positive contribution to the velvet deer germplam resources in China. English text needs a  careful revision.

However i have some difficult to understand some sentences. Wky the author writes livestock and poulty ??; poultry (??), lines 23.; Lines 38-39, are not well written.

Lines 109, have been cultivated??; This manuscript described animal genetic resources,  namely velvet deer germplam. Why the author writes  about seed production. It confounds any reader.  Lines 146-150, are not clear;  

See, also in conclusions, lines 258-260, are not clear. It talks again livestock and poultry germplam, seed industry. 

So moderate revision are needed, before acceptance  for publication.

Author Response

Comments:

  1. “Why the author writes livestock and poulty ??; poultry (??), lines 23. ;”

Response:

Thank you for your valuable suggestion. Our description herein is not accurate. As you have said, this article has nothing to do with poultry germplasm resources. The sentence has been rewritten in the revised manuscript in Line 23.

  1. “Lines 109, have been cultivated??; This manuscript described animal genetic resources, namely velvet deer germplam. Why the author writes about seed production. It confounds any reader.”

Response:

The paragraph where line 109 is located is a description of the germplasm resources of Chinese domestic red deer. Firstly, this paragraph describes the living domestic red deer germplasm resources. And then, line 109 emphasizes that Cervus elaphus yarkandensis has formed a cultivated breed called Tahe Red Deer, it is a kind of domestic red deer germplasm resource. We also made a correction in lines 121-123.

  1. “See, also in conclusions, lines 258-260, are not clear. It talks again livestock and poultry germplam, seed industry.”

Response:

The lines 258-260 is an overview of the status of poultry and livestock germplasm resources in China. “Seed industry” is not used accurately, we have modified the sentence in lines 305-307.

  1. “Lines 38-39, are not well written.; Lines 146-150, are not clear and lines 258-260, are not clear.”

Response:

    Lines 38-39 have been rewritten in lines 40-43; lines 146-150 have been rewritten in lines 160-162; lines258-260 have been rewritten in lines 305-311.

We are sorry to make you feel difficult to understand the paper. The language of this paper has been appropriately modified.

Special thanks to you for your good comments and have a good day.

Reviewer 2 Report

Current situation and utilization of velvet deer germplasm resources in China

Lixin Tang, Xiaobin Wen, Ranran Zhang and Xiumei Xin

This paper gives a useful summary of the status and location of Chinese sika and red deer subspecies and breeds, and summarises some of the information on genes relevant to the growth of velvet antler. The purpose of the paper is to describe the cervine gene pool of China and to indicate how this information might be used to halt the decline in the Chinese velvet industry. They authors say (lines 38-40) that “scientific protection and rational utilization of velvet deer germplasm resources will be the key to solving the current velvet deer industry crisis and promoting the stable development of velvet deer industry”. While the paper identifies these issues, it would be strengthened by a more detailed discussion of how present and proposed actions could improve velvet production.

The paper should be edited by a proficient English-language speaker, so as to ensure that the authors’ meaning is correctly understood. Some of the difficulty relates to the use of undefined terms which might not be understood by non-Chinese readers.

There are several points which I think the authors should consider:

·         line 23: “Recent years, the provenance protection and innovation have been paid attention to”. (1) This sentence is unclear – what do you mean by “provenance” in this context? (2) What and how is the germplasm protected, and what innovations are you referring to? (3) How have these things been “paid attention to”?

·         lines 37-38: “With the emphasis on the seed industry and a series of measures to revitalize the seed industry.” What is the “seed industry”; is it breeding pure-bred animals, similar to the breeding of pure-bred bulls and cows on beef stud farms?

·         line 47: “China is rich in deer resources, with 21 species, accounting for 41.7% of the world [1].” 41.7% must refer to the number of deer species and subspecies in China, rather than the number of animals. Please clarify this.

·         lines 91-93: “Velvet deer have been domesticated in China for hundreds of years, and C. n. hortulorum is the most dominant species in domestic sika deer. At present, the domestic sika deer in China is mainly from C. n. hortulorum.”  These two sentences are repetitive.

·         lines 93-97: “Based on this, one local variety (Jilin sika deer), seven breeding varieties (Shuangyang sika deer, Xifeng sika deer, Aodong sika deer, Siping sika deer, Xingkai lake sika deer, Dongda sika deer and Dongfeng sika deer) as well as one breeding strain (Changbai mountain sika deer) identified by the National Livestock and Poultry Breed Committee have been bred.” (1) Please explain what local and breeding varieties are and how these differ, and the difference between a breeding variety and a breeding strain. (2) How were these varieties and strains bred, and what relevance did this exercise have to the topic of this paper?

·         lines 165-166: “meat production performance such as meat ratio, slaughter rate and net meat rate”. Please define these terms.

·         line 211: “and used as a reference genome for GWAS analysis”. Please define GWAS.

·         lines 221-224: “The improved variety of velvet deer requires not only pure-blooded, superior production performance and the shape of pilose antler, but also the unique appearance characteristics of the variety. Therefore, multi-dimensional identification and evaluation should be conducted when identifying the germplasm resources of velvet deer. (1) Does “pure-blooded” mean “pure-bred”, i.e. animals which are not hybrids or crosses between breeds? (2) Why does high quality antler have to be produced from “pure-blooded” animals; is there any evidence for this? (3) By multi-dimensional identification and evaluation, do you mean that studies of the genomes of different deer species should try to identify genes which will influence antler shape (characteristic of the deer species) rather than concentrating only on production characteristics?

·         lines 253-255: “the genomic selection reference group of improved varieties need to be established, the breeding chip for high-density SNP and breeding value evaluation model need to be developed”. (1) Please explain in more detail what a “genomic selection reference group of improved varieties” would constitute and describe how this grouping could be used to promote velvet antler production. (2) What is a “breeding chip”?

·         Table 1. “the distribution and number of the subspecies wild sika deer.” This is very useful information. It would also be useful to include data on the current Red List status of these species where that is available.

·         lines 224-230: “Firstly, the appearance and the production performance of velvet deer were evaluated according to National Standards for Chinese Sika Deer and National Standards for Northeast Red Deer. And then, the purity of the velvet deer was detected by genome sequencing technology. Finally, the genetic resource data center of velvet deer was established and the velvet deer that were identified as improved varieties were registered, which would provide the basis for the follow-up work on germplasm resources protection and breeding innovation.” This description is in the section entitled “Suggestions on conservation and utilization of velvet deer germplasm resources” and appears to describe events that have occurred in the past. It’s not clear why this material is included in a section on suggested future actions. Please rewrite the paragraph to clarify whether these actions are suggestions for the future, or, if they have already been done, how the results of these actions have aided the Chinese velvet industry.

·         lines 232 – 234: “Breeding innovation with a stable genetic basis must be based on pure-blooded, so it is necessary to establish a sound protection system of velvet deer germplasm resources for the healthy development of the velvet deer industry.” Do you mean by “protection” that the existing deer subspecies should be prevented from cross-breeding (except under controlled conditions)?

·         lines 234-243: “In order to better implement the protection project of velvet deer germplasm resources, we must base on the local improved variety resources, strengthen the cooperation between research departments and enterprises, build the national velvet deer improved variety protection farm and breeding farm relying on the existing large and medium-sized breeding bases, and carry out purification and rejuvenation work using gene detection technology. In addition, we will expand the gene bank of improved varieties, vigorously do research on the germplasm characteristics of velvet deer genetic resources, conduct evaluation and analysis, dig out genes related to production traits and promote the development and utilization of local varieties, so that indirectly achieve the purpose of conservation.” (1) These objectives are all aspirational; can you describe how these things are being done at present, or how they might be done in the future? (2) You reiterate the idea that the best velvet antler would be produced by pure-bred animals, is this assumption been tested and found to be correct? (3) “Dig out” is very informal usage; “identify” would be a better choice.

Author Response

  1. Response to comment: “(lines 38-40) that “scientific protection and rational utilization of velvet deer germplasm resources will be the key to solving the current velvet deer industry crisis and promoting the stable development of velvet deer industry”. While the paper identifies these issues, it would be strengthened by a more detailed discussion of how present and proposed actions could improve velvet production.”

Response:

As you said the improvement of velvet production is also a problem for velvet deer industry. However, we emphasize that nowadays the crisis of velvet deer industry is the misuse of velvet deer germplasm resources in this manuscript and the detailed discussion is shown in the suggestion part. We have corrected the sentence in lines 40-43.

  1. line 23: “Recent years, the provenance protection and innovation have been paid attention to”. (1) This sentence is unclear – what do you mean by “provenance” in this context? (2) What and how is the germplasm protected, and what innovations are you referring to? (3) How have these things been “paid attention to”?

Response:

We are sorry for using the wrong term. We have modified this sentence in lines 23-25.

  1. lines 37-38: “With the emphasis on the seed industry and a series of measures to revitalize the seed industry.” What is the “seed industry”; is it breeding pure-bred animals, similar to the breeding of pure-bred bulls and cows on beef stud farms?

Response:

We have rewritten this sentence for “It reminds us that scientific protection and rational utilization of velvet deer germplasm resources will be the key to solving the problem about the misuse of germplasm resources and promoting the stable development of velvet deer industry” in lines 40-43.

 Seed industry is a general name of a series of industries that work for the investigation, collection, sorting and preservation of germplasm resources, breeding of new varieties and the testing and promotion of varieties [1].

[1] Song, L-N; Wu, D-W; Hou, J-Q. Research on the Development and Implementation Strategy of Seed Industry in China. Journal of Northwest AF University(Social Science Edition), 2022, 22(06):141-149. doi:10.13968/j.cnki.1009-9107.2022.06.16.

  1. line 47: “China is rich in deer resources, with 21 species, accounting for 41.7% of the world [1].” 41.7% must refer to the number of deer species and subspecies in China, rather than the number of animals. Please clarify this.

Response:

We have clarified this sentence to “China is rich in deer resources, with 21 species, accounting for 41.7% of deer species in the world” in line 50.

  1. lines 91-93: “Velvet deer have been domesticated in China for hundreds of years, and C. n. hortulorum is the most dominant species in domestic sika deer. At present, the domestic sika deer in China is mainly from C. n. hortulorum.” These two sentences are repetitive.

Response:

We have modified the sentence to “Velvet deer have been domesticated in China for hundreds of years. At present, the domestic sika deer in China is mainly domesticated from C. n. hortulorum.” in lines 101-103.

  1. lines 93-97: “Based on this, one local variety (Jilin sika deer), seven breeding varieties (Shuangyang sika deer, Xifeng sika deer, Aodong sika deer, Siping sika deer, Xingkai lake sika deer, Dongda sika deer and Dongfeng sika deer) as well as one breeding strain (Changbai mountain sika deer) identified by the National Livestock and Poultry Breed Committee have been bred.” (1) Please explain what local and breeding varieties are and how these differ, and the difference between a breeding variety and a breeding strain. (2) How were these varieties and strains bred, and what relevance did this exercise have to the topic of this paper?

Response:

I would like to answer your question. Firstly, Local varieties refer to wild species that are domesticated artificially and have not been intensively selected for breeding, but still retain rich genetic diversity. However, breeding varieties refer to varieties with relatively stable heredity formed by at least four generations of strict breeding standards on the basis of local varieties. Local varieties are the basis for cultivating breeding varieties. Secondly, breeding varieties refer to varieties with relatively stable heredity formed by at least four generations of strict breeding standards on the basis of local varieties. And breeding strain need two generations of breeding. The difference between breeding varieties and strain is the breeding time. Finally, the purpose of this paragraph is to introduce the germplasm resources of domestic sika deer in China and the local variety, seven cultivated varieties and a cultivated strain are the available domestic deer germplasm resources in China. Our focus is not to introduce the breeding process of these germplasm resources. We have also change “breeding varieties” to “cultivated varieties” and “breeding strain” to “cultivated strain” in lines 101-106.

  1. lines 165-166: “meat production performance such as meat ratio, slaughter rate and net meat rate”. Please define these terms.

Response:

We are very sorry for our incorrect writing about these terms. We have corrected these terms and added a reference in lines 180-181.

  1. line 211: “and used as a reference genome for GWAS analysis”. Please define GWAS.

Response:

Thanks for your comment, we have added the full name of GWAS in line 228.

  1. lines 221-224: “The improved variety of velvet deer requires not only pure-blooded, superior production performance and the shape of pilose antler, but also the unique appearance characteristics of the variety. Therefore, multi-dimensional identification and evaluation should be conducted when identifying the germplasm resources of velvet deer. (1) Does “pure-blooded” mean “pure-bred”, i.e. animals which are not hybrids or crosses between breeds? (2) Why does high quality antler have to be produced from “pure-blooded” animals; is there any evidence for this? (3) By multi-dimensional identification and evaluation, do you mean that studies of the genomes of different deer species should try to identify genes which will influence antler shape (characteristic of the deer species) rather than concentrating only on production characteristics?

Response:

We are sorry to confuse you because of our inaccurate description. We have changed “pure-blooded” to “pure-bred” and rewritten the sentence in lines 239-240.

 We do not want to express that high quality antler have to be produced from “pure-brred. Pure-bred animal has a clean genetic background, it is a precious material and foundation for breeding innovation. we think it is a rational way to identify and evaluate the germplasm resources of velvet deer from Multiple aspects. As you suggested that evaluating by genomes research is pretty good.

  1. lines 253-255: “the genomic selection reference group of improved varieties need to be established, the breeding chip for high-density SNP and breeding value evaluation model need to be developed”. (1) Please explain in more detail what a “genomic selection reference group of improved varieties” would constitute and describe how this grouping could be used to promote velvet antler production. (2) What is a “breeding chip”?

Response:

The genomic selection reference group is a reference population that uses in the breeding of genomic selection. Genomic selection is based on the data of reference groups with genotype and phenotype to build models and predict the phenotypic value of only genotype individuals [2].

Breeding chip is a chip that gathers a large number of SNPs that are significantly related to the productive performance, disease resistance, stress resistance and other dominant traits of species. These SNPs which are significantly related to dominant traits were obtained by genome-wide association study of the species population with dominant traits in the previous study.

We have rewritten this part in lines 285-295.

[2]Grevenhof I E V; Werf J H V D. Design of reference populations for genomic selection in crossbreeding programs. Genetics Selection Evolution, 2015, 47(1):1-9. DOI: 10.1186/s12711-015-0104-x

  1. Table 1. “the distribution and number of the subspecies wild sika deer.” This is very useful information. It would also be useful to include data on the current Red List status of these species where that is available.

Response:

Thanks for your comments. Sika Deer has most recently been assessed for The IUCN Red List of Threatened Species in 2014 and Cervus nippon is listed as Least Concern. However, the wild populations of sika deer were listed as the first-class national protected animal in the latest released version of The Lists of Wild Animals under Special State Protection in China (WASPC) in 2021. We have added a reference in line72-74 and line 76-79.

  1. lines 224-230: “Firstly, the appearance and the production performance of velvet deer were evaluated according to National Standards for Chinese Sika Deer and National Standards for Northeast Red Deer. And then, the purity of the velvet deer was detected by genome sequencing technology. Finally, the genetic resource data center of velvet deer was established and the velvet deer that were identified as improved varieties were registered, which would provide the basis for the follow-up work on germplasm resources protection and breeding innovation.” This description is in the section entitled “Suggestions on conservation and utilization of velvet deer germplasm resources” and appears to describe events that have occurred in the past. It’s not clear why this material is included in a section on suggested future actions. Please rewrite the paragraph to clarify whether these actions are suggestions for the future, or, if they have already been done, how the results of these actions have aided the Chinese velvet industry.

Response:

This description is the details that we need to follow for identifying and evaluating the velvet deer germplasm resources scientifically. We have modified this part in lines 239-250.

  1. lines 232 – 234: “Breeding innovation with a stable genetic basis must be based on pure-blooded, so it is necessary to establish a sound protection system of velvet deer germplasm resources for the healthy development of the velvet deer industry.” Do you mean by “protection” that the existing deer subspecies should be prevented from cross-breeding (except under controlled conditions)?

Response:

We want to express that pure-bred has a clean genetic background and it is the basis of breeding innovation. However, cross-breeding is an effective way of breeding innovation. Therefore, it is important to keep the balance between the protection of pure-bred resources and the development of breeding innovation. It is necessary to establish a sound system to protect the velvet deer germplasm resources as well as make innovations in cross-breeding to the greatest extent.

We have modified this sentence in lines 252-257.

  1. lines 234-243: “In order to better implement the protection project of velvet deer germplasm resources, we must base on the local improved variety resources, strengthen the cooperation between research departments and enterprises, build the national velvet deer improved variety protection farm and breeding farm relying on the existing large and medium-sized breeding bases, and carry out purification and rejuvenation work using gene detection technology. In addition, we will expand the gene bank of improved varieties, vigorously do research on the germplasm characteristics of velvet deer genetic resources, conduct evaluation and analysis, dig out genes related to production traits and promote the development and utilization of local varieties, so that indirectly achieve the purpose of conservation.” (1) These objectives are all aspirational; can you describe how these things are being done at present, or how they might be done in the future? (2) You reiterate the idea that the best velvet antler would be produced by pure-bred animals, is this assumption been tested and found to be correct? (3) “Dig out” is very informal usage; “identify” would be a better choice.

Response:    

As you suggested, we have rewritten the sentences in lines 258-272. We think Pure-bred velvet deer has a clean genetic background and it is a precious material and foundation for breeding innovation.

Special thanks to you for your valuable comments and have a good mood.

Reviewer 3 Report

Review of animals-2012693 entitled “Current situation and utilization of velvet deer germplasm resources in China” by Tang et al.

General comment: The paper is clearly written and shows that the velvet deer faces the problems of germplasm resources protection and international competition and how the breeding of new varieties besides genetic improvement of existing deer varieties will develop in high-quality germplasm resources.

I believe those suggestions on conservation are of great importance for utilizing velvet deer germplasm resources in China.  

Specific comments:

Line 251: 4.3 starts in line 252.

Line 269: There are two consecutive dots. Remove one.

The bibliography does not follow a particular reference pattern. It is incomplete, with missing doi in some references (39, 41, ....) and wrong doi in others such as 36 (should be DOI: 10.1093/genetics/144.4.1331).

Author Response

1.Line 251: 4.3 starts in line 252.

We have corrected the line 252.

2.Line 269: There are two consecutive dots. Remove one.

We have removed one consecutive dots in line 269.

3.The bibliography does not follow a particular reference pattern. It is incomplete, with missing doi in some references (39, 41, ....) and wrong doi in others such as 36 (should be DOI: 10.1093/genetics/144.4.1331).

Thank you for your valuable suggestion. We have corrected the doi of all reference and completed some missed doi. However, there are still some references (11,15,16,38,41,42) that we cannot find their doi from the website of China DOI (http://www.chinadoi.cn/).

Sincere thanks for your valuable comments and have a good mood.

Reviewer 4 Report

The authors developed an interesting study about determination of "Current situation and utilization of velvet deer germplasm resources in China". Review focus on the situation and utilization of velvet deer germplasm resources in China. Introduced the situation of velvet deer germplasm resources in China, such as ecological distribution and the number of subspecies. Described the domestication and breeding development of velvet deer in China and put forward suggestions on the scientific utilization of velvet deer germplasm resources. In my opinion, the presented manuscript is of practical and protect value.

The bibliographic list should be corrected in accordance with the requirements of the journal.

I believe that the manuscript corresponds to the requirements of the journal and therefore I recommend this article for publication.

Author Response

  1. The bibliographic list should be corrected in accordance with the requirements of the journal.

As you suggested, we have modified all the references by referring to reference style of the published article.

Special thanks for your approval and valuable comment of the article.

Have a good mood.

Reviewer 5 Report

File attached

Author Response

  1. Line 47 Sika deer and red deer are two species belonging to the same Cervus (???) do you mean ‘same genus’?

As you suggested, we have corrected the “Cervus” to “genus” in line 48.

  1. Line 58 Please correct、 in , (also row 59)

We have changed “、” to “,” in lines 59-60.

  1. Line 60-63 Due to the change……………….. sinchuanicus remain [1]. Non clear, please, rephrase

We have rewritten the sentence to “Due to the change in environmental climate and the interference of many human factors, the number of wild sika deer has dramatically decreased. The existing subspecies of sika deer are Cervus nippon hortulorum, Cervus nippon kopshi, Cervus nippon taiouanus and Cervus nippon sinchuanicus.” in lines 60-63.

  1. Line 68-69 Edit phrase

We have modified the sentence to “The wild population of Cervus nippon taiouanus was recovered from extinction through artificial rearing. The number of wild individuals were about 1000 by 2013 and mainly lived in Taiwan” in lines 68-70.

  1. Line 82 most subspeciesof red

We have corrected “subspeciesof” to “subspecies of” in line 82.

  1. Line 83 Cervus nippon taiouanus Cervus nippon taiouanus please modify all the nomenclature

Thank you for your valuable suggestion, we have modified all the nomenclature of the manuscript.

  1. Line 85 Red deer are is

We have changed “are” to “is” in line 85.

  1. Line 89 Populus euphratica see line 83

We have rewritten the sentence in lines 88-89.

  1. Line 92 Please draft table 2, it is unnecessarily wide and it is awkward to read

We have readjusted the width of the table 2 in line 92

  1. Line 94 has

We have changed “have” to “has” in line 96.

  1. Line 96 cultivated varietes (are they vegetables???) please write ‘reared subspecies‘

Rear means to care for young children or animals until they are fully grown or to breed or keep animals or birds, for example on a farm. However, we want to express “varieties have been bred” in this sentence. We have corrected “cultivated varieties” to “bred varieties” in line 98.

  1. Line 98 cultivated reared

Rear means to care for young children or animals until they are fully grown or to breed or keep animals or birds, for example on a farm. However, we want to express “a strain has been bred” in this sentence. So we have corrected “cultivated strain” to “bred strain” in line 100.

  1. Line 112 cultivated reared please remove all cultivated that follow

We have corrected all “cultivated” to “bred”.

  1. Line 114-6 Please rephrase

We have rephrased the sentence in lines 116-118.

  1. Line 124 7331 frozen semen samples

We have corrected this sentence in lines 124-126.

  1. Line 142 At the same time, in the 1960s, the average yield (of what?) of Shuangyang sika………..

We have corrected this sentence to “In the 1960s, the average velvet antler yield of Shuangyang sika

deer exceeded 1kg, ranking first in antler production in China” in lines 143-144.

  1. Line 149 to cultivate see above

We have corrected “cultivate” to “breed” in line 149.

  1. Line 163 cultivate

We have corrected “cultivate” to “breed” in line 164.

19.Line165-6 F1 can breed normally, and its reproductive performance, antler-producing performance, growth performance ……….please rephrase

We have rewritten the sentence in lines 165-169.

  1. Line 187 used foridentifying

We have changed “foridentifying” to “for identifying” in line 186.

  1. Line 245 the cost of construction (???)

We have corrected the sentence to “reduce the cost of conservation farm construction” in lines 242.

  1. Line 287 Sheng, H-L > Sheng, H.L. Please check accurately all references

Thank you for your valuable suggestions. We have corrected all references by referring to the published article of Animals.

  1. Line 374 Daguin‐Thiébaut

We have corrected the name in line 383

Great thanks for your valuable suggestions. The language of the manuscript has been appropriately modified.  Have a good day.

Round 2

Reviewer 2 Report

Thankyou for dealing with the points raised in my previous review. I think that the paper is acceptable for publication.

Author Response

Special thanks to you for your approval and have a good mood.

Reviewer 5 Report

Dear Authors,

            I appreciate that the suggestions have been largely taken into account, especially in favour of greater clarity in the overall presentation of the text.

Best regards

Round 3

Reviewer 5 Report

Dear Authors,

            I appreciate that the suggestions have been largely taken into account, especially in favour of greater clarity in the overall presentation of the text.

Best regards